# ‘Tracking Together’—Simultaneous Use of Human and Dog Activity Trackers: Protocol for a Factorial, Randomized Controlled Pilot Trial

**DOI:** 10.3390/ijerph18041561

**Published:** 2021-02-07

**Authors:** Wasantha Jayawardene, Lesa Huber, Jimmy McDonnell, Laurel Curran, Sarah Larson, Stephanie Dickinson, Xiwei Chen, Erika Pena, Aletha Carson, Jeanne Johnston

**Affiliations:** 1Department of Applied Health Science, School of Public Health, Indiana University Bloomington, Bloomington, IN 47405, USA; lehuber@indiana.edu (L.H.); set9@indiana.edu (S.L.); 2Department of Kinesiology, School of Public Health, Indiana University Bloomington, Bloomington, IN 47405, USA; jimcdon@iu.edu (J.M.); lscurran@iu.edu (L.C.); penae@iu.edu (E.P.); jdjohnst@indiana.edu (J.J.); 3Department of Epidemiology and Biostatistics, School of Public Health, Indiana University Bloomington, Bloomington, IN 47405, USA; sd3@indiana.edu (S.D.); xiwechen@indiana.edu (X.C.); 4Data and Clinical Research, Kinship Division, Mars Petcare, Inc., 18101 SE 6th Way, Vancouver, WA 98683, USA; aletha.carson@effem.com

**Keywords:** human activity trackers, canine activity trackers, dog walking

## Abstract

Dog-walkers are more likely to achieve moderate-intensity physical activity. Linking the use of activity trackers with dog-walking may be beneficial both in terms of improving the targeted behavior and increasing the likelihood of sustained use. This manuscript aims to describe the protocol of a pilot study which intends to examine the effects of simultaneous use of activity trackers by humans and their dogs on the physical activity level of humans and dogs. This study uses nonprobability sampling of dog owners of age 25–65 (*N* = 80) and involves four parallel groups in an observational randomized controlled trial with a 2 × 2 factorial design, based on use of dog or human activity trackers for eight weeks. Each group consists of dog-human duos, in which both, either or none are wearing an activity tracker for eight weeks. At baseline and end, all human subjects wear ActiGraph accelerometers that quantify physical activity for one week. Commercial activity trackers are used for tracking human and dog activity remotely. Additional measures for humans are body composition and self-reported physical activity. Dog owners also report dog’s weight and physical activity using a questionnaire. A factorial analysis of covariance (ANCOVA) is used to compare physical activity across the four groups from baseline to week-10.

## 1. Introduction

While physical activity is a vital component of health and well-being of not only humans, but also of dogs, physical inactivity is identified as a leading contributor to chronic diseases such as obesity, diabetes, and heart disease in both humans and dogs [1,2,3,4]. Rising levels of chronic diseases have been linked to physical inactivity in humans and dogs that continue to challenge length and quality of life for both dogs and their owners [3,4,5,6]. To date, numerous human studies have demonstrated that short periods of moderate-to-vigorous intensity activity, and even regular, moderately-paced walking, such as dog-walking, can be beneficial for disease prevention and control [7,8]. A recent review has revealed that dog-walkers are more than 2.5 times more likely to achieve moderate-intensity physical activity, defined as at least 150 min per week [9]. Numerous studies reveal that dog ownership is often associated with increased walking time and leisure-time physical activity overall among adults [10,11,12] as well as a heightened sense of community [13]. Although evidence on the effects of dog walking on dog health is relatively scarce, obesity is now the top most health concern in dogs globally [14]. Studies reported that the risk of being overweight among dogs declined gradually for each one hour of exercise undertaken [15].

According to U.S. pet ownership statistics from 2016, 38% of households had at least one dog, while dogs continued to remain the most popular pets in households, with a total population of pet dogs of approximately 77 million, up 10% from 2011 [16]. Therefore, promotion of dog walking can be a broad but realistic public health intervention [17], which can be achieved by increasing the potential for dog walking through implementing population-level policies, environments, and programs. In addition to owning a dog, several other factors are appeared to impact daily dog walking behavior: dog’s size medium or large was positively associated but owning multiple dogs and having a greater number of persons in the household were negatively associated. Empirical studies have revealed that many dog owners lack motivation or experience barriers to walking their dogs [12,18,19,20,21].

Activity trackers have been shown to have a positive impact on human physical activity [22,23,24]. Increases in daily steps, more intense physical activity, and expenditure of energy are potential benefits of using these trackers [25]. Although activity trackers (i.e., accelerometers) have the potential for improving activity behaviors [26], activity trackers do not appear to encourage sustained use, as many adults no longer use activity trackers after about six months [27]. However, linking the use of activity trackers with a targeted behavior that has demonstrated positive health effects, such as dog-walking [28], may be beneficial both in terms of improving the targeted behavior and increasing the likelihood of sustained use. Dog activity trackers, while relatively new, have demonstrated promise in monitoring dog physiological indicators [29]. There are dog activity monitoring devices that collect data describing dog physical activity, inactivity, sleep, and other behaviors. So, combining the use of activity trackers by dog owners along with dog activity trackers could be particularly effective in creating sustainable exercise routines, especially dog-walking.

Preliminary research suggests that dog activity trackers may increase bonding between dogs and their owners and increase awareness of the dog’s behavior and well-being as well [30]. However, what is not known is the effect of simultaneous paired use of human and dog activity trackers on physical activity. Given that dog owners are more likely to achieve suggested weekly physical activity, simultaneous use by owners and dogs could lead to an even larger increase in physical activity. A collocated human–dog experience may be personally compelling and sufficiently positive to promote sustainable motivation and happier, healthier, longer lives for both dogs and humans [31]. The ability of owners to track not only their personal activity, but also the health of their dog, may increase commitment and dedication to dog-walking, subsequently leading to increased physical activity levels.

The research question of this study is: What impact does the simultaneous use of activity trackers by humans and their dogs have on the physical activity level of humans and dogs? The objective of this manuscript is to describe the methods and protocols of an ongoing 2 × 2 factorial randomized controlled trial which intends to examine the effects of simultaneous use of activity trackers by humans and their dogs on the physical activity level of humans and dogs.

## 2. Materials and Methods

### 2.1. Study Setting

The study is being conducted in Bloomington, Indiana, United States. It is a college town that has 85,000 population. About 99% of the population is urban. The racial composition is 83.0% white alone, 8.0% Asian alone, 4.6% black alone, and 3.5% Hispanic. Median household income is $42,000, and the median age is 23.4 years.

### 2.2. Eligibility Criteria

Inclusion criteria for human participants are (1) owning a dog, (2) age between 25 and 65 years, and (3) owning a Fitbit- and Whistle-compatible smartphone, because an app is required to use activity trackers. Exclusion criteria for human participants are (1) those who are participating in another study that requires physical activity or exercise, (2) those who have substantial physical activity limitations according to a self-reported screening procedure (see Subjective Measures below), (3) those who are already wearing activity trackers, and (4) those who are not the primary caretaker of a dog (e.g., those who walk another person’s dog). Dogs are not brought to laboratory for measurements, so eligibility criteria for dogs are determined based on input from dog owners. Dogs of any breed or age who do not have physical activity restrictions or disability are eligible to participate.

### 2.3. Study Design

This pilot study uses convenience (nonprobability) sampling to draw a sample from the town’s residents who are easy to contact or to reach. The study involves four parallel groups (Figure 1), in a purely observational randomized controlled trial with a 2 × 2 factorial design based on use of dog or human activity trackers for eight weeks. Fitbit Charge-3 is used for tracking human physical activity (physical activity duration, steps, distance walked). The dog activity tracker “Whistle Fit” is used for tracking of dog activity (physical activity duration, distance walked). Each group consists of dog–human duos, in which none, either, or both is/are wearing an activity tracker for eight weeks. Thus, there are 4 groups: (1) neither dogs nor humans wear a tracker; (2) only dog owners wear activity trackers; (3) only dogs wear activity trackers; (4) both dogs and owners wear activity trackers. The trial is being conducted during December 2020 through June 2021. Outcomes are measured both continuously under free-living conditions and at two time points in a kinesiology lab—at baseline and end-of-trial. Additionally, before (pre-randomization) and after the 8-week period in groups, all human subjects wear a research-grade accelerometer that quantifies baseline and end-of-trial human physical activity, respectively, for one week (week-1 and week-10). The study is considered observational because researchers do not ask study participants to change their physical activity level or any other behavior, rather use activity trackers to observe any behavioral changes associated with participants’ activity tracker use. Considering the purely observational nature, this study does not require clinical trial registration.

### 2.4. Sample Size Calculation

A sample size of 20 per group (80 total) provides 80% power to detect a “medium/large” effect size (Cohen’s f = 0.32) for main effects (dog tracker; owner tracker) or the interaction. These effect sizes are consistent with previous literature on increase in physical activity after using pedometers [32]. While 20 per group has limited power in the case of more subtle (i.e., smaller) effects, the means in each group are estimated within 0.46 standard deviations of the mean so that means and SD can be reliably used as pilot data in larger future studies. The percentage of estimated dropout is 20%, so for intent-to-treat analysis (see Analysis Plan), this study aims to recruit and randomize 100 human-canine duos (25 per group).

### 2.5. Recruitment

This study involves a staggered recruitment process that allows to recruit participants in small batches. Flyers, advertisements on notice boards in public places (e.g., parks), and campus-wide email invitations are used to inform potential participants about the study. Advertisements are also be sent to kennels, veterinarians, and dog trainers. An email address and phone number are included for potential participants to contact a research investigator. Direct mail/email, flyers/brochures, and websites are used to contact potential subjects. Subject self-referral in response to recruitment materials are the only source of participants for the study. Advertisements state that, to take part in the study or for more information, potential participants (those who are interested in participating) have to contact the research team directly by email or go to the website where they can submit their contact information. A research assistant, who is a kinesiology graduate student involved in the project, follows up by phone with the potential participant.

### 2.6. Objective Measures

Before (pre-randomization) and after the 8-week period in groups (Figure 1), all human subjects wear an GT3X-BT ActiGraph™ (ActiGraph™, Pensacola, FL, United States) research-grade tri-axial accelerometer that quantify human physical activity for one week (week-1 and week-10). For this, in the beginning of week-1, subjects come to the lab and pick up the accelerometer. Subjects wear this on the left hip during waking hours, except during bathing or water activities, for seven consecutive days. As per subject choice, either paper or electronic forms are provided for subjects to keep track of the accelerometer wear time. Physical activity, including daily walking steps, sedentary time, and intensity of physical activity are measured using accelerometers. Total, moderate, and vigorous physical activity are calculated for each subject. After one week of wearing the accelerometer they come back to the lab to return the accelerometer. During this meeting, subjects are informed what group they are assigned to (see Randomization). If it is the Fitbit, Whistle or Fitbit plus Whistle, they receive those devices then (Fitbit Charge-3 for dog owners or Whistle-Fit for dogs or both). In the next phase of the study which lasts eight weeks, human participants and their dogs, except for those in true control group, wear activity trackers, based on their group assignment. During this 8-week period, human activity trackers in Group-2 and Group-4 provide visual feedback to the human participants regarding patterns of activity and allow for the setting of goals for themselves. Dog activity trackers in Group-3 and Group-4 provide visual feedback to the human participants regarding patterns of their dog’s activity and allow for the setting of goals for their dogs. Once participants synchronize activity trackers with smartphone app, research team is able to extract these data remotely. In the end of this 8-week period in groups, subjects come to lab again for measurements. All human subjects, including those who were in the true control group, receive an ActiGraph accelerometer again to wear for an additional week (week-10). Dog owners are not asked to bring dogs to lab for measurements, instead they are requested to enter dog’s weight in the baseline questionnaire, using information from the most recent visit to veterinarian.

Additional objective measures for humans are body composition with height and weight (for BMI), waist circumference, and bioelectrical impedance analysis (BIA), which are measured by a research assistant in a secure room. The scales are calibrated and set to zero before all measurement procedures. Height is measured to the nearest 0.1 cm via a stadiometer (Charder Height Measurement) in a standing position with shoes removed, shoulders relaxed, facing forward with back facing the wall. Height is taken twice and averaged for reliability and accuracy. Weight is recorded, using a digital standing scale, to the nearest 0.1 kg with minimal clothing on. Waist circumference is measured to the nearest 0.1 cm at the level of the iliac crest while the subject is at minimal respiration. Waist circumference is measured three times and averaged for reliability and accuracy. BIA is performed using a Tanita MC-780U. The BIA produces readings on body weight, body fat, body muscle, BMI, and basal metabolic rate in kcals. Participants are instructed to abstain from intense exercise, food, and beverage for four hours prior to the BIA.

### 2.7. Subjective Measures

Physical limitations that would exclude participants from the study are determined by the potential participants’ responses to the validated Physical Activity Readiness Questionnaire (PAR-Q) [33] that is administered as a screening procedure. Human physical activity is measured using validated International Physical Activity Questionnaire (IPAQ) [34]. A reasoned action approach (RAA) framework is used to explore beliefs and determinants of dog-walking behavior [35]. Dogs’ physical activity is measured using a non-validated questionnaire, developed for the purpose of this study, based on validated study instruments [10].

### 2.8. Randomization

After all participants complete the run-in phase in the week-1 with the ActiGraph GT3X-BT accelerometer, they turn in their device and are randomized to one of the four groups for whether the dog or the owner receives the activity tracker: (1) neither, (2) only the owner, (3) only the dog, (4) both the dog and owner. The biostatistics team created a randomization schedule using block-randomization in SAS software and provided the study team with sealed envelopes for concealed allocation. However, as this is a behavioral trial, blinding is impossible after random assignment.

### 2.9. Compensation

At the end of the study, all participants, including those in the true control group, receive a Fitbit Charge-3 and a Whistle Fit. Those participants that used a Fitbit or a Whistle Fit during the study keep those devices. Those participants who did not use these devices during the study receive these devices. To receive these devices, participants are required to wear an ActiGraph accelerometer for one week each in the beginning and the end of the study.

### 2.10. Ethical Considerations

The human subjects research protocol was approved by the Institutional Review Board (IRB) of Indiana University Bloomington, United States. Animal research protocol was approved by the Animal Care and Use Committee (ACUC) of the same institution. Informed consent document and verbal script approved by IRB of the research institution are used. All subjects receive the informed consent document by email and provide verbal informed consent during a scheduled phone call. During this call, the research assistants (kinesiology graduate student) explain the study to the potential subject verbally, providing all pertinent information, study purpose, and procedures, and allow the potential subject ample opportunity to ask questions. There are no immediate physical, psychological, social, legal, and economic risks associated with this study. Furthermore, dog-walking itself may lead to unexpected events, such as road traffic accidents, falls, etc., for both humans and dogs. Finally, a security breach in activity tracker usage data can cause a threat to confidentiality. Research team members who are measuring height, weight, waist circumference, and BIA, and administering questionnaires are properly trained for human subject research and handling relevant equipment, so unexpected events (e.g., accidents) are highly unlikely. Regarding questionnaires, participants are asked to skip any questions they do not want to answer. Furthermore, participants are given a phone number and email address of two research team members to contact if they have questions about any risk related to research activities. In the meantime, participants and their dogs in the intervention groups receive several benefits, i.e., they have the opportunity to increase their physical activity, which may lead to positive health outcomes, such as improved cardiovascular, physical, and mental health.

All participant data collected, retained, and protected by the research team are identifiable. This is essential for the follow-up of participants. Each participant who is enrolled and randomized in the study are assigned a unique anonymous identification number that is linked to their personal identification, and only designated study staff have access to personal identification for contacting participants. This way, the participants’ data can still be collected and linked across visits, but individually identifiable information does not become part of the research record or dataset that are stored and used for analysis. Only deidentified data are available to data analysts. Data retrieved from human activity trackers are collected and organized in text (.txt) files or Excel spreadsheets. Data retrieved from human activity trackers, canine activity trackers, and Qualtrics questionnaires are organized separate text files or spreadsheets. Participants’ data are not shared with third parties, unless required by State or Federal law; this is informed to participants at recruitment. Deidentified data necessary to reproduce data analyses in publications will be made available to the public. Appropriate data safety monitoring (DSM) protocols are set up to ensure safety and privacy of participants. Data collected during the study are entered on Excel by a graduate student at the data collection site at the Indiana University Bloomington. Access to the computer is protected with a password. No confidential data are stored in personal computers or at homes. Deidentified data are then stored on a secure Microsoft OneDrive, maintained by Indiana University Bloomington, for transfer and use by the biostatisticians for summary reports and analyses. Frequent data back-up systems are used to prevent loss of data. Confidential paper files will be stored under lock and key in the Principal Investigator’s office for a three-year period following project termination; then will be securely discarded.

### 2.11. Data Management

Electronic Data Capture will be used for Qualtrics online survey instruments and electronic data from Fitbit and Whistle activity trackers, so that no data are transcribed by study staff. During the data collection process, Indiana University-affiliated research staff will monitor missing or invalid data and send reminders to wear and synchronize activity trackers. Range checks and valid data fields will be implemented in survey instruments. After the data are collected, outliers, duplicate data and invalid data will be investigated, and data analysis will perform full diagnostic tests for influential data points.

### 2.12. Analysis Plan

For the physical activity analysis, the following variables will be extracted or calculated from summary spreadsheets: (1) total sedentary time and percentage; (2) low, light, moderate, and vigorous physical activity time, and percentage; (3) total moderate to vigorous physical activity (MVPA) time and percentage; and (4) average daily step count. Daily averages will be calculated by dividing the values of these variables by the number of days the participant wore the ActiGraph GT3X-BT accelerometer. A 2 × 2 factorial analysis of covariance (ANCOVA) will be used to compare physical activity across the 4 groups from baseline to week-10. Fixed effects are dog tracker (Yes/No), human tracker (Yes/No), and interactions. The primary outcome of physical activity will be obtained from ActiGraph GT3X-BT accelerometer data in the week-1 run-in phase and the week-10 following the 8-week period in groups, measured as change. Physical activity will be analyzed as total, moderate, and vigorous physical activity. Covariates will be included for baseline outcomes as well as demographics of the dog (age, sex, breed, size, activity level from questionnaire) and the owner (age, sex, BMI, activity level from questionnaire). Although this is a pilot study, sensitivity analyses will be conducted with subgroups for season of the year, and variables from the behavior questionnaires will be utilized to interpret results. While small sample sizes preclude large models, small subsets of demographic variables will be included based on their effect sizes. Descriptive summaries will be generated to examine any potential imbalances between the groups. To account for missing data after attrition, linear mixed models will also be performed to include all participants randomized (intent-to-treat) where partial-data will be included on people with baseline data even if missing follow-up data.

### 2.13. The Impacts of Pandemic 

The clinical trial experienced substantial delays due to COVID-19 (Coronavirus Disease 2019) pandemic. Subsequently, researchers decided which outcome measures and time points are critical to achieve the research objectives and which protocol procedures can be accurately and safely completed during the pandemic [36]. Accordingly, original protocols were revised to mitigate the risks associated with COVID-19, while attempting to maximize evidence acquisition; for example, paper and pencil questionnaires in laboratory were transformed into Qualtrics surveys that can be administered remotely. Collection of imperfect data was preferred to non-collection, for example, highly accurate dual energy X-ray absorptiometry (DEXA) for body composition was replaced with less accurate BIA, whereas an additional timepoint of data collection (week-5) for all outcome measures was dropped. The U.S. Centers for Disease Control and Prevention (CDC) guidelines were followed for COVID-19 risk mitigation plans and protocols, such as moving the study data collection site from the School’s main building to a small building on campus, screening each study participant for symptoms and potential COVID-19 exposures prior to each lab visit, reducing the interaction between each graduate student and each study participant to a duration that does not exceed 15 min, and adhering to standard social distancing and disinfecting guidelines. Precautions were also taken to minimize potential difficulties with interpretability of results and statistical analysis due to missing data. All protocol amendments and deviations have been approved by the IRB of the research institution.

## 3. Results

The data collection of this study is ongoing. This study has two hypotheses: (1) The physical activity level of humans is higher with simultaneous use of activity trackers by humans and their dogs compared with no tracker use and human-only use; (2) The physical activity level of dogs is higher with simultaneous use of activity trackers by humans and their dogs compared with no tracker use and dog-only use. Descriptive statistics and the results of hypotheses testing will be published in an upcoming manuscript. See Data Availability Statement below for details pertaining to sharing of deidentified individual participant data (IPD) following the publication of final results.

## 4. Conclusions

Linking the use of activity trackers with dog-walking may be beneficial both in terms of improving the targeted behavior and increasing the likelihood of sustained use. According to empirical literature to date, this is the first randomized controlled trial that used the factorial design to study dog-walking behavior and associated outcomes in both species. Considering the increasing prevalence of dog-ownership and emerging technologies for pet activity tracking, more research is needed to evaluate the effects of simultaneous use of activity trackers by humans and dogs on both species. The results of this study will provide insights into future interventions that promote physical activity.

## Figures and Tables

**Figure 1 ijerph-18-01561-f001:**
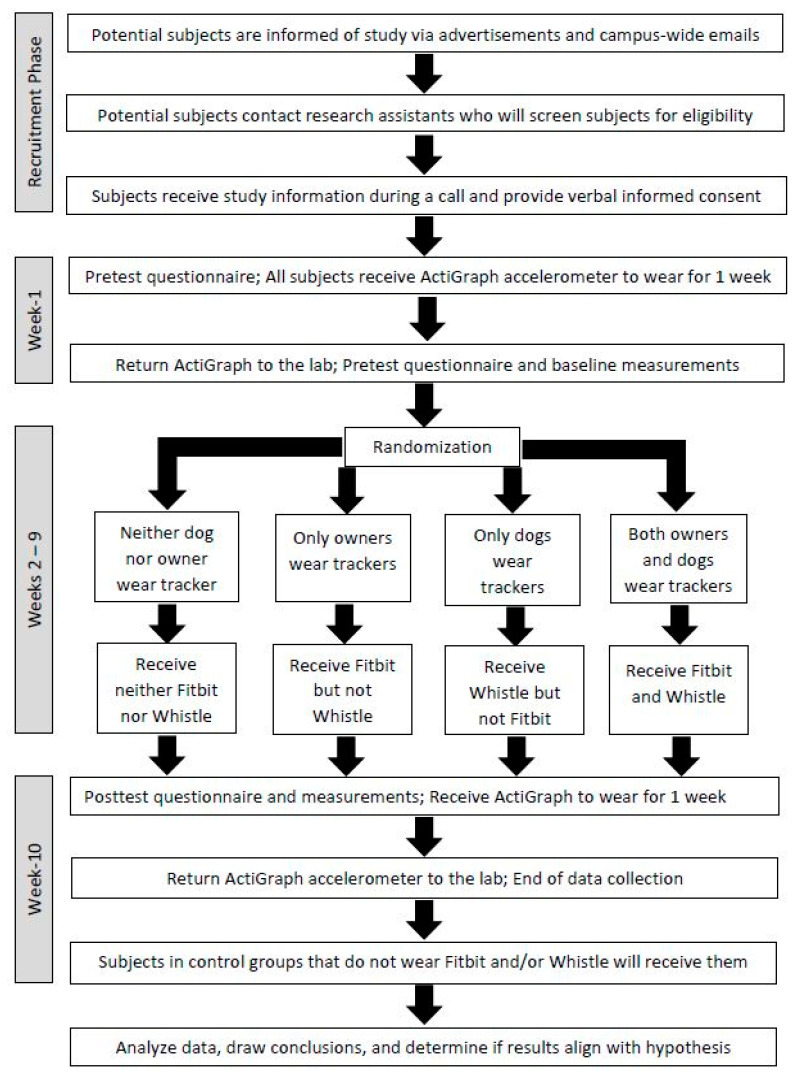
Flow Chart for 2 × 2 Factorial Randomized Trial on Use of Human and Dog Activity Trackers.

## Data Availability

The data collected in this study will be deidentified to make available to other researchers. Deidentified individual participant data (IPD) will be available from the corresponding author on formal request, beginning six months and ending five years following the date of publication of final results. The data will not be publicly available due to ethical considerations.

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
