# Peer review of "‘Tracking Together’—Simultaneous Use of Human and Dog Activity Trackers: Protocol for a Factorial, Randomized Controlled Pilot Trial"

_ijerph, 2021, doi:10.3390/ijerph18041561_

Round 1
Reviewer 1 Report
Comments:
- Line 18-20: Authors are advised to revise the sentence for the clarity.
- Abstract: It’s quite interesting, however cannot find any results or conclusions in the Abstract that are essential to understand.
- Line 41-42: Please mention the specific age or age group of people in those studies 10, 11 and 12.
- References should cite uniformly. For example Lines 70-84, there are both numbers and author names for the references. Please check the whole manuscript for similar corrections.
- Although authors provided informative background, its bit of too long. Authors may shorten the Introduction.
- Line 187-189: Check and revise the sentences for clarity.
- Is there any source or supporting reference for the dog’s questionnaire?
- Subsection 2.9. Compensation: Is it necessary to include in the paper?
- Authors stated that this study approved by IRB, however no approval number was provided.
- Authors well described about the study design and participants. It would be nice if authors can provide further details including male/female numbers, average age of human subjects in each group, age and bodyweights of dogs and other basic details.
- Authors mentioned that data will be published in an upcoming manuscript. So what is the strength or finding of this study and what authors would like to tell to the readers?
- If protocol or methodology is the major contribution, authors at least could provide the baseline characteristics of participants and dogs.
- Does the methods given in this study need reliability and validity assessment?
- What are the expected outcomes from this study?
Reviewer 2 Report
I have added my comments in the file attached.

Reviewer 3 Report
In my opinion, the paper has a low scientific impact, but from the point of view of the structure it is impeccable. I consider that it can be a guide for future research.
In my opinion, the research is interesting, and its results should be made known to the scientific community, but what the authors want is that the study protocol be published. In my opinion, it does not have the scientific relevance to be published since it does not propose a complex methodology or something original. What it is basically about is using an electronic device to assess levels of physical activity. This research tries to measure the physical activity of dogs and the people who walk them. and above all to evaluate the effect of the different feedback on the amount of physical activity carried out (dog-human). The use of electronic devices, including accelerometers, is very common in the scientific literature.
In my opinion, a relevant aspect is the sample: it has important limitations. I think it has many biases. On the one hand the groups are very limited in relation to the number of subjects, in my opinion, 20 subject are very few, especially considering the age range from 25-65. That is, we can have subjects of different ages and sexes, which causes the groups to be very heterogeneous. Another variable is the breed of the dog, one is more active and others are more sedentary. (e.g) the Border Collie is very active and the Great Dane is very sedentary. I think the inclusion and exclusion criteria should be reviewed to make the groups homogeneous.
Another determinate aspect that is not sufficiently clarified in the method when the test will be carried out, it is said that between December and June. It must be taken into account that the weather is decisive in the practice of physical activity, especially in dog walking. Specifically, the trial will take place in an area of the USA where temperatures are extreme, very cold in winter and very hot in summer.
In the research protocol, the results section must be more defined. And in the discussion it is not correct, in my opinion, to address the issue of COVID. 19. The information in the discussion should be in the method.
The results of this research may be interesting and help to understand the motivation to carry out activities.
Round 2
Reviewer 1 Report
The authors addressed all the comments. The revised manuscript is suitable for publication.
Reviewer 3 Report
The authors have made the modifications that I have suggested